# Surgical Treatment of Glioblastoma: State-of-the-Art and Future Trends

**DOI:** 10.3390/jcm11185354

**Published:** 2022-09-13

**Authors:** Arthur H. A. Sales, Jürgen Beck, Oliver Schnell, Christian Fung, Bernhard Meyer, Jens Gempt

**Affiliations:** 1Department of Neurosurgery, Klinikum rechts der Isar, Technical University of Munich, Ismaninger Str. 22, 81675 Munich, Germany; 2Department of Neurosurgery, Universitätsklinikum Freiburg, 79106 Freiburg, Germany

**Keywords:** glioblastoma, surgery, extent of resection, residual tumor volume, intraoperative magnetic resonance imaging, intraoperative fluorescence

## Abstract

Glioblastoma (GBM) is a highly aggressive disease and is associated with poor prognosis despite treatment advances in recent years. Surgical resection of tumor remains the main therapeutic option when approaching these patients, especially when combined with adjuvant radiochemotherapy. In the present study, we conducted a comprehensive literature review on the state-of-the-art and future trends of the surgical treatment of GBM, emphasizing topics that have been the object of recent study.

## 1. Introduction

Glioblastoma (GBM) is the most common and malignant primary brain tumor in adults with a 5-year mortality rate > 90% [1,2]. Annually, more than 10,000 cases are reported in the United States [3].

For many decades, the standard treatment of GBM consisted of adjuvant radiotherapy following surgical resection [4]. However, since 2005, the introduction of adjuvant temozolomide combined with postoperative radiotherapy became the new standard treatment and improved the median survival of these patients [5,6].

Maximum safe resection remains a fundamental part of this treatment and represents the main objective when surgically approaching these patients due to its association with longer survival [7].

In order to achieve this goal, new techniques and surgical adjuncts (e.g., fluorescence-guided surgery, intraoperative magnetic resonance imaging (iMRI), brain mapping strategies, intraoperative ultrasound (IOUS), confocal intraoperative microscope (CIM), and intraoperative mass spectrometry (IMS)) have been investigated and employed over the last few years. Intraoperative radiotherapy (IORT) may reduce the incidence of local recurrence and prolong survival. However, the results of clinical studies are still conflicting. Experimental methods such as Raman spectrometry (RS) and optical coherence tomography (OCT) have shown promising results in small experimental and clinical studies. Nevertheless, there is still a lack of high-quality evidence on these topics.

In the present article, we offer a comprehensive literature review on the state-of-the-art and future trends of the surgical treatment of GBM.

## 2. Extent of Resection and Residual Tumor Volume: Agreements and Controversies

The extent of resection (EOR) is one of the most investigated topics regarding the surgical treatment of GBM. On the one hand, it has been already demonstrated that EOR affects the overall survival (OS) of patients with GBM; on the other hand, there has been much debate about the optimal threshold of the EOR. The highly cited paper by Lacroix et al. described that an EOR ≥ 98% improved median survival from 8.8 months (95% confidence interval (CI) 7.4–10.2) to 13 months (95% CI 11.4–14.6), *p* < 0.0001 (see Table 1) [8]. In addition, they reported a stronger association between the EOR and survival when other prognostic factors such as age and Karnofsky Performance Score (KPS) were favorable. However, care must be taken when interpreting the results, since newly diagnosed and recurrent GBMs were not evaluated separately. Orringer et al. retrospectively evaluated 46 patients with newly diagnosed GBM and concluded that an EOR greater than 90% was significantly associated with greater 1-year survival [9].

Oppenlander et al. studied 170 patients with recurrent GBM and reported improved OS in patients with EOR ≥ 80% [11]. They emphasized the fact that patients with EOR ≥ 80% showed a higher risk of developing neurological morbidity in the early postoperative period than patients with EOR < 80% [11]. However, this increased risk did not last beyond 30 days [11]. Another study with 500 patients with newly diagnosed GBM showed similar results regarding the impact of EOR on OS [10]. Even though greater EOR was associated with higher survival rates, the role of subtotal resection (STR) in patients with GBM was demonstrated, since benefits were seen with as little as 78% EOR [10]. This is particularly important in patients with tumors located adjacent to or within eloquent areas, where achieving EOR ≥ 98% may not be possible. However, a study involving 345 patients with newly diagnosed GBM showed that complete tumor resection correlated with significantly improved survival (HR: 0.6, *p* = 0.003), while patients who underwent incomplete resection did not show a longer OS than those who received needle biopsy [24].

Bloch et al. demonstrated that gross total resection (GTR) at recurrence is associated with improved survival regardless of initial EOR [12]. They reported that patients with initial STR had improved survival (15.9 months to 19 months, *p* = 0.004) when receiving GTR at recurrence [12]. A meta-analysis that included 1507 patients, of whom 1335 had recurrent tumor, described that maximal resection at recurrence was significantly associated with improved survival [13].

Another meta-analysis studied 1618 patients of three retrospective and three randomized controlled trials and concluded that GTR was associated with greater 1-year OS and PFS when compared with STR [15].

The role of residual volume (RV) in the surgical treatment of GBM has also been investigated in many previous studies. Grabowski et al. reported that RV was the most significant predictor of survival compared with EOR, T2/FLAIR residual volume, and contrast-enhanced preoperative tumor volume [16].

Chaichana et al. evaluated the association between RV and EOR with the survival of patients with newly diagnosed GBM and established minimum EOR and maximum RV thresholds. They reported that the minimum EOR threshold for survival and recurrence was 70%, while the maximum RV threshold for survival and recurrence was 5 cm^3^ [18]. In another retrospective study, Chaichana et al. evaluated 84 patients with newly diagnosed GBM who were considered amenable to GTR based on preoperative imaging. RV and EOR were independently associated with survival. In addition, they reported that the RV and EOR with the greatest impact on OS was <2 cm^3^ and >95%, respectively [14].

Bette and colleagues retrospectively studied 209 patients with newly diagnosed GBM and confirmed that surgical resection remains a major prognostic factor, since RV remained significantly associated with survival even after adjusting the model for other prognostic factors such as age, KPS, MGMT-status, and adjuvant radiochemotherapy [17].

A retrospective study investigated which parameter is more important for the prognosis of newly diagnosed GBM: RV or EOR [20]. The authors reported that regardless of STR or GTR, EOR was not significantly associated with OS and PFS, in contrast to RV, which showed potential to provide greater predictive power for the prognosis of GBM [20]. These results were confirmed by another retrospective review of 147 GBM patients that demonstrated a significant association between RV < 3.5 cc and survival of patients who received incomplete tumor resection [19]. A significant association between EOR and survival could not be demonstrated in this study [19]. In addition, another retrospective study of 64 patients with recurrent GBM reported that RV, but not EOR, showed prognostic power in both univariate and multivariate analyses [21].

Orringer et al. demonstrated the influence of tumor location on EOR. They reported that EOR was less for tumor located in eloquent areas and those touching ventricles [9]. It is well known that aggressive resection of these tumors may increase the risk of postoperative neurological morbidity. Therefore, neurosurgeons may be more conservative when operating on these lesions, resulting in lesser EOR.

The role of molecular features in the surgical treatment of GBM has also been investigated. A retrospective study evaluated 126 patients with MGMT-unmethylated GBM and revealed that complete tumor resection was not associated with improved survival. However, they emphasized that maximum safe resection should always be attempted, since RV is significantly associated with OS [22].

Shah et al. explored the influence of supramaximal resection or anatomic lobectomy on the survival of patients with non-eloquent gliomas. Their propensity-matched analysis showed that supramaximal resection resulted in improved OS (30.7 vs. 14.1 months) and PFS (17.2 vs. 8.1 months) compared to the GTR group (*p* < 0.001) [25]. Another study reported that the subpial technique extended tumor resection beyond the contrast enhancement and is associated with longer OS compared to similar series where resection of contrast-enhanced tumor was performed [23]. Contrastingly, a retrospective study demonstrated that the EOR of FLAIR-hyperintense areas did not improve the survival of patients with GBM [26]. Prospective randomized studies are necessary in order to further investigate this topic.

The role of EOR and RV in the surgical treatment of GBM has been the subject of debate in the field of neurosurgery for years. The conflicting results presented in this review reflect the different methodologies of the presented studies on the one side and the heterogeneity of examined populations on the other side. It is important to emphasize, however, that there is a consensus with respect to the principle of maximum safe resection. The golden rule regarding the extent of resection of GBM in the year 2022 is: to resect as much contrast-enhanced tumor as possible without causing neurological deterioration. The level of evidence provided by the medical literature on this topic is limited due to the fact that most published studies are based on retrospective analyses. New prospective randomized studies are needed to address this important issue related to the surgical treatment of patients with GBM.

## 3. Fluorescence-Guided Surgery: An Indispensable Innovation

Given the fact that greater EOR and lesser RV improve the OS and PFS of GBM patients, the development of new techniques intending to improve resection rates without causing neurological morbidity is necessary. 5-aminolevulinic acid (5-ALA), a natural precursor of hemoglobin, is a fluorescent dye that is preferably picked up by tumor cells after being orally administered 2–3 h prior to surgery [7,27]. See Figure 1.

A modified neurosurgical microscope can visualize the fluorescence originating from tumors cells, thus improving EOR [27]. In this context, Stummer et al. conducted a multicenter randomized controlled trial with 322 patients with suspected malignant glioma in order to investigate whether 5-ALA-induced fluorescence had a significant impact on EOR and 6-month progression-free survival, as assessed by MR images (Table 2) [27]. GTR was achieved in 65% of patients in the 5-ALA group in comparison with 36% in the white light group (*p* < 0.0001) [27]. Moreover, patients assigned to the 5-ALA group had higher 6-month PFS (41%) than those in the white light group (21.1%), *p* = 0.0003 [27]. The incidence of neurological deterioration did not differ significantly between groups 7 days after surgery (18% 5-ALA group vs. 10% control group, *p* = 0.2) or 6 weeks after surgery (17% 5-ALA group vs. 12% control group, *p* = 0.3) [27]. This clinical trial provided high-level evidence regarding the benefits of using 5-ALA-guided resection in patients with malignant gliomas. Another study with 36 GBM patients analyzed the efficacy of 5-ALA-guided resection and reported complete resection of the contrast-enhanced lesion in 83% of cases, EOR over 98% in 100% of cases, and mean EOR of 99.8% [28]. In addition, it was demonstrated that strong fluorescence identified solid tumor with 100% positive predictive value based on histopathological and immunohistochemical analyses of biopsies with different fluorescence intensities [28].

A meta-analysis of 20 studies including a total of 565 patients who underwent 5-ALA-guided resection reported a mean GTR rate of 75.4%, mean time to tumor progression of 8.1 months, and mean overall survival gain of 6.2 months [29]. Additionally, the evaluation of 800 histological samples showed a sensitivity of 82.6% (95%CI: 73.9–91.9, *p* < 0.001) and specificity of 88.9% (95%CI: 83.9–93.9, *p* < 0.001) [29]. Another meta-analysis reported similar results regarding sensitivity (87%) and specificity (89%) [34].

When compared with iMRI, 5-ALA had higher sensitivity and specificity for detecting tumor infiltration at the border of the resection cavity in patients with high-grade gliomas (HGGs) [35]. The impact of the combined use of 5-ALA-guided resection and iMRI on EOR was demonstrated in a prospective study with 33 GBM patients eligible for GTR [36]. In this group of patients, a combined approach with 5-ALA and iMRI was performed. The control group was selected through a retrospective matched pair assessment in 144 patients with iMRI-assisted surgery. Mean EOR was significantly higher in the combined therapy group (99.7%) than in the iMRI-alone group (97.4%), *p* < 0.004. Additionally, the rate of GTR was significantly higher in the combined therapy group (100% vs. 82%, *p* < 0.01) [36]. Another study with 72 patients with GBM demonstrated that higher rates of GTR can be achieved when 5-ALA-guided resection is combined with intraoperative monopolar mapping in tumors located adjacent to the corticospinal tract [37].

Aldave et al. investigated whether the presence of residual fluorescent tissue in patients with GTR as confirmed by postoperative MRI had a significant impact on OS and neurological complication rate. The median OS was significantly higher in patients with GTR and no residual fluorescent tissue (27.0 months, CI= 22.4–31.6) than in those with GTR and residual fluorescent tissue (17.5 months, CI= 12.5–22.5), *p* = 0.015 [30]. Age, tumor volume, and 18F-FET PET uptake are predicting factors for 5-ALA fluorescence in tumors without typical GBM radiological features [38,39].

In patients with recurrent GBM, 5-ALA is also a valid adjunct tool, although care must be taken when trying to differentiate reactive tissue changes caused by adjuvant radiochemotherapy from true disease progression [40]. In addition, 5-ALA is considered a useful adjunct during iMRI-guided resection of malignant gliomas, since it allows identification of tumor tissue beyond its radiological borders [41]. Roder et al. compared EOR, RV, and neurological outcomes of patients who underwent GBM surgery with 5-ALA, iMRI, or conventional white light and reported better results in the group of patients in the iMRI group [42].

5-ALA-guided surgery is associated with high sensitivity and specificity for identifying malignant tumor tissues and represents an intraoperative tool independent of neuronavigation for achieving maximal EOR without causing neurological deterioration [31,32,33,43,44]. Additionally, this surgical adjunct seems to be cost-effective in comparison with conventional white light surgery in patients with HGG [45,46]. Thereby, it has become an indispensable innovation in the surgical treatment of GBM.

Fluorescein is also used as a fluorescent-tracer, and it accumulates in areas where the blood–brain barrier is damaged [47].

In neurosurgery, it was first used in Japan in the late 1990s in a series of 30 patients and showed promising results [48]. A prospective study evaluated the influence of fluorescein-guided surgery on GTR and survival in a series of 80 patients with GBM. The authors reported a significantly higher rate of GTR in patients who had fluorescein-guided surgery than those who had conventional surgery (83 vs. 55%). However, survival did not differ between the two groups [49]. Other studies confirmed that fluorescein sodium is safe and allows a high rate of complete removal of contrast-enhanced tumor [47,50].

The use of tumor-targeted molecular imaging, in the form of near-infrared (NIR) fluorescent dyes, can improve detection, margin control, and survival in many cancer subtypes [51]. Recently, Miller et al. published the first-in-human study with cetuximab-IRDye800, an antibody against the epidermal growth factor receptor (EGFR). They reported that this method presented feasibility and safety in patients with GBM [51].

Intraoperative MRI: Technological Advance versus Medical Evidence

Intraoperative MRI has been used as an adjunct tool in the context of glioma surgery since 1993, when it was first introduced by the Brigham and Women´s Hospital [52].

It is well known that the brain shift phenomenon reduces the accuracy of conventional neuronavigation during resection of brain tumors. Therefore, the main idea behind this innovative technique is the possibility of improving EOR by means of intraoperative updated MR images. This real-time assessment allows the possibility of further resection in the same surgery [53]. However, current evidence supporting this practice is still limited by the fact that most studies on iMRI are retrospective cohorts and case–control studies [52].

The first randomized controlled trial on this topic was conducted by Senft et al. and enrolled 58 adult patients with contrast-enhanced gliomas for which GTR was planned (Table 3). Patients in the iMRI group had a higher rate of GTR (96%) than those in the control group (68%), *p* = 0.023. Moreover, the incidence of postoperative neurological deficits did not differ between groups (13% in the iMRI group vs. 8% in the control group), *p* = 1.0 [54]. An important limitation of this study was the fact that neurosurgeons were not blinded to treatment allocation, which may lead to treatment bias. Additionally, an ultra-low-field MRI device was used, which provides inferior images resolution in comparison with high-field devices. Lastly, the sample size did not allow subgroup analysis of different glioma grades.

A recent prospective, triple-blind, controlled trial analyzed 87 patients randomly assigned to either the iMRI group or control group. The rate of GTR was significantly higher in the iMRI group (86.36%) than in the control group (53.49%), *p* < 0.001 [52]. The benefit of 3.0-T iMRI-guided resection in improving EOR was significant for LGG (*p* = 0.01), and there was a slight, but non-significant trend for HGG (*p* = 0.2). Furthermore, no significant difference was found regarding the occurrence of postoperative neurological deficits between treatment groups [52]. PFS analysis as estimated by Kaplan–Meier curves indicated a trend toward improved 6-month PFS of patients with GBM in the iMRI group (*p* = 0.24) [52]. Another study involving 200 patients with HGG has shown that iMRI had no significant impact on OS after adjusting baseline discrepancies in preoperative KPS [55].

In another study, 135 patients with GBM underwent 1.5T-iMRI-guided resection. Tumor remnant was found in 88 patients of whom 19 underwent extended resection. In 9 of these 19 patients, further resection resulted in GTR, which represented an increase in the GTR rate from 34.80% to 41.49% [56].

A prospective study involving 14 patients investigated the importance of combining 5-ALA-fluorescence- and iMRI-guided resection in glioblastoma surgery. After complete resection of 5-ALA fluorescent tissue, iMRI was performed in order to identify areas with suspicion of remnant tumor [61]. These suspicious lesions underwent biopsy and were sent to histopathological evaluation. iMRI showed areas suspicious for tumors in 91.6% of cases after complete resection of 5-ALA fluorescent tissue [61]. However, histopathological confirmation of remnant tumor occurred in only 64.3% of cases [61]. This fact highlights the low predictive value of iMRI for identifying tumor remnants, which has to be considered when performing extended resection of contrast-enhanced lesions near eloquent areas.

In another study, patients who were 5-ALA fluorescence-negative had better resection rates when undergoing combined iMRI-guided resection (89.2%) than those without iMRI (68.7%) [62]. Coburger et al. evaluated a series of 170 surgeries for GBM with iMRI and concluded that surgery in a contemporary setup using iMRI and standard adjuvant treatment presented higher OS and lower complication rates, as previously published [57]. On the other hand, Kubben et al. published an interim analysis of a randomized trial on iMRI-guided GBM surgery compared to conventional neuronavigation and reported no advantage with respect to EOR, clinical performance, and survival in the iMRI-guided group [58]. However, the low statistical power due to the small sample size (14 patients) may have led to a type 2 error (false negative result). A retrospective study including 114 patients who underwent GBM surgery reported that the use of iMRI enhanced both EOR (overall GTR: 88.5% vs. 44%) and 6-month PFS (73% vs. 38.9%) [59]. Moreover, a meta-analysis of RCTs and retrospective studies described a positive impact of iMRI on 6-month PFS and the rate of GTR, but no difference in EOR, tumors’ size reduction, or time required for surgery between the two neuronavigation approaches [60].

Although it has been shown that iMRI increases EOR in patients with brain tumors, there is a lack of high-level evidence supporting that the use of this technological advance results in significant improvement in PFS, OS, and quality of life. Furthermore, iMRI is associated with a longer operation time and implies the use of appropriate surgical instruments, which may lead to higher treatment costs [63].

New randomized controlled trials with a larger sample size and long-term follow-up are needed in order to investigate whether iMRI provides significant survival benefits to patients with GBM.

## 4. Intraoperative Ultrasound: A Widely Available and Inexpensive Tool

Intraoperative ultrasound (IOUS) is considered a widely available and inexpensive adjunct tool in the surgical treatment of brain tumors. Its benefits have been demonstrated by several studies in the last few decades [64]. In the early 1990s, Le Roux et al. investigated the impact of IOUS on the identification of tumor margins in 33 patients with low-grade glioma (LGG). They reported that 85% of patients who were eligible for complete resection (20 patients) had ultrasound-defined margins that were free of solid tumors in the histopathological analysis. Therefore, it was concluded that IOUS may enhance delineation and EOR in patients with LGG [65]. Another study from the 1990s showed that IOUS improved the identification of tumor margins beyond falsely underestimated margins defined by preoperative T1 images [66]. Solheim et al. observed that there is an important association between image quality and clinical and radiological results. They suggested that better ultrasound promotes better surgery [67]. In addition, a prospective study with 88 patients undergoing surgical resection of glioma detailed that the use of navigable ultrasound was associated with significantly better PFS and OS (Table 4) [68]. Another study retrospectively analyzed 192 GBM patients and reported an improvement in survival rates within the period that IOUS and neuronavigation were introduced and established in their department (Figure 2) [69].

The influence of a 3D-ultrasonography-based navigation system on the quality of life (QOL) of 88 patients with glioma was investigated by Jakola et al. They suggested that the use of IOUS may be associated with a preservation of QOL in these patients [70]. In addition, another study concluded that navigable 3D US is a versatile, useful, and reliable intraoperative tool in the surgical treatment of brain tumors [74].

A meta-analysis of 15 studies including 739 glioma patients showed that the use of IOUS was associated with improved EOR, especially when the lesion was solitary and subcortical, with no history of surgery or radiotherapy [71]. Prada et al. investigated the role of contrast-enhanced ultrasound (CEUS) in the identification of residual tumor mass in 10 patients undergoing GBM surgery. The study detailed that in one case only, CEUS partially failed to demonstrate residual tumor. Consequently, it was concluded that CEUS is extremely specific in the identification of residual tumor mass [72].

Another study compared the impact of navigated versus non-navigated IOUS on the EOR of patients with HGG and reported no difference regarding tumor remnant sizes between groups [75]. A recent study demonstrated that navigated intraoperative ultrasound improved EOR and neurological outcomes when compared to standard neuronavigation [73].

It is important to mention that IOUS is a user-dependent tool. Therefore, the knowledge, skills, and experience of the neurosurgeon play a decisive role in the utility of this adjunct tool in the surgical treatment of GBM. Official diplomas and certification might be offered by national and international neurosurgical societies in order to ensure teaching standards, skill requirements, and revalidation practices [76].

Although many studies on this topic have been published in recent years, high-quality evidence is still desirable. Further research will improve the usefulness of this method in GBM surgery.

## 5. Intraoperative Radiotherapy: Targeting Infiltrative Margins

GBM is a highly aggressive disease, and despite the use of adjuvant radiochemotherapy, local recurrence near the resection cavity remains a major clinical issue and is associated with clinical deterioration and death [77].

The interval between surgical resection of the brain tumor and the beginning of adjuvant treatments may have a crucial impact on local recurrence due to the presence of remaining cells in the resection cavity and its margins [78]. Therefore, it was hypothesized that the application of IORT could reduce the local recurrence of glial tumors (Figure 3).

In this context, Japanese scientists conducted the first studies indicating a benefit of using intraoperative radiotherapy in patients with malignant gliomas in the early 1990s (Table 5) [79,80,81]. At that time, IORT was mainly delivered by means of electron-based intraoperative radiotherapy (IOERT), which has several technical limitations, including inadequate electron cone sizes, inadequate low energies, and areas of insufficient target volume coverage [77]. Sakai et al. conducted a clinical study involving 73 patients with glioblastoma and anaplastic astrocytoma and reported that IORT improved median survival from 20.7 months to 26.2 months (*p* < 0.01) [79]. In another study, 20 or 25 Gy of irradiation was delivered in a single fraction in 20 of 36 glioma patients. Median survival time of the IORT group was 14 months vs. 10 months in the control group [80]. On the other hand, Schueller et al. related that the OS of 71 patients treated with IORT (20–25 Gy) was not improved compared to a historical control group. This study also described that rates of postoperative complications were not increased in patients who had received IORT [82]. Other small studies from the 1990s demonstrated the efficacy and relative safety of IORT in the treatment of malignant brain tumors [83,84,85]. A recent open-label, dose-escalation phase I/II trial reported that the use of IORT (20 to 40Gy of low-energy photons) is associated with manageable toxicity [86]. The main adverse events reported were: radionecrosis, wound dehiscence, CSF leakage, and cyst formation [86]. In addition, a retrospective study reported that dose escalation with IORT was associated with significantly lower healthy brain exposure in comparison to preoperative stereotactic radiosurgery [87]. Intraoperative radiosurgery also allowed higher dose delivery to the surgical bed [87].

Usychkin et al. analyzed 17 patients with newly diagnosed malignant gliomas and 15 patients with recurrent tumors who had received intra-operative electron beam radiotherapy (IOERT). The median overall survival was 14 months and 10.4 months for the primary and recurrent cohort, respectively. The study concluded that IOERT is a feasible technique and may be used as a tool in the treatment of malignant gliomas [90].

An international pooled analysis of 51 patients suggested the improved efficacy and safety of IORT with low-energy X-rays for newly diagnosed GBM [89]. A clinical study involving 32 patients who were treated with IORT (12–15 Gy) followed by external radiation therapy (median dose 60 Gy) reported no survival benefit compared to matched control cases [88]. Much is expected from the use of radioenhancers and radiosensitizers, which can increase the radiation delivery while sparing the surrounding normal brain tissue [91].

The results of clinical studies regarding the use of IORT in patients with GBM are still conflicting to date. This may be explained by the use of different methodologies in previously published studies (e.g., radiation doses, IORT techniques). Moreover, most available literature is based on non-randomized small studies, which do not provide a high level of medical evidence. New prospective randomized trials are warranted in order to evaluate the real impact of IORT on the survival outcomes of patients with GBM.

## 6. Brain Mapping, Monitoring Strategies, and Awake Surgery: Locating and Preserving Critical Functions

While adjunct tools such as fluorescence-guided surgery, neuronavigation, and iMRI attempt to technically simplify maximal tumor resection, the safety of these procedures is mostly provided by brain mapping devices and techniques, especially for tumors located near or within so-called eloquent areas. The role of these surgical adjuncts in the context of GBM surgery has been becoming more important in the last few decades, and much research has been devoted to this field. Consequently, tumors considered non-resectable years ago are now eligible for resection with an acceptable rate of postoperative morbidity [92]. The main objective of brain mapping is to determine which anatomical areas are responsible for critical functions (e.g., motor and language) in order to preserve them during surgical procedures. It is well known that tumor cells infiltrate areas beyond the contrast-enhanced lesion. Therefore, functional-guided surgery using brain mapping and monitoring strategies may overtake and replace imaging-guided surgery, especially in the context of supratotal resection.

Didactically, brain mapping techniques can be divided into preoperative mapping (navigated transcranial magnetic stimulation (nTMS), functional MRI (fMRI), magnetoencephalography (MEG), and diffusion tensor imaging fiber tracking (DTI-FT)) and intraoperative mapping (direct cortical stimulation (DCS) and subcortical stimulation (SCS)) [93]. See Figure 4.

nTMS has been shown to be a suitable and reliable technique for preoperative surgical planning in patients with brain tumors. A study involving 14 patients with tumors located close to the precentral gyrus mentioned a high correlation between nTMS and DCS (gold standard) in locating the motor cortex. In addition, the attending neurosurgeons declared that the intraoperative identification of the central region was simplified by using this technique [94]. A systematic review of 11 publications reported that nTMS correlated well with DCS in all included studies. Moreover, based on the data of 87 patients of two studies, they also reported that surgical strategy was changed in 25.3% of cases, which emphasizes the relevant role of nTMS in helping neurosurgeons in the decision-making process [95].

Some studies demonstrated that nTMS improves the outcome of patients with motor eloquent lesions [96,97]. Krieg et al. related that patients who were preoperatively investigated by nTMS had a significantly lower rate of postoperative residual tumor volume on MRI, a higher rate of postoperative clinical improvement, a lower rate of neurological deterioration, and smaller craniotomies than those without preoperative nTMS [97]. Another matched case–control study has shown that the use of preoperative nTMS led to a significant increase in the GTR rate from 42% to 59% (*p* < 0.05), improved PFS from 15.4 months to 22.4 months (*p* < 0.05), and helped neurosurgeons in the surgical planning. This study also mentioned a non-significant reduction in the postoperative deficit rate from 8.5% to 6.1% [96]. 

Language mapping with repetitive nTMS is still under investigation in many neurosurgical departments around the world. Recent studies, however, demonstrated a high correlation with DCS [98,99,100]. Furthermore, nTMS has been also used to investigate the phenomenon of neuroplasticity before and after surgical removal of tumors located in eloquent areas [101,102,103].

The role of fMRI as an efficient tool for preoperative brain mapping remains controversial (Figure 5) [104,105,106,107,108]. Even though some studies have found a high correlation with intraoperative cortical mapping and high efficacy for locating motor and language areas, other studies have failed to demonstrate such an association [104,105,106,107].

In addition, false-negative activation on fMRI can occur due to vascular changes, which might lead to iatrogenic resection of eloquent tissues and, consequently, to postoperative morbidity [93,109].

While fMRI provides information based on hemodynamic parameters, MEG directly measures cortical activity through the detection of magnetic fields generated during neuronal activation [110]. Schiffbauer et al. related that preoperative MEG correlated well with DCS in patients with brain tumors and may be considered an accurate adjunct for preoperative mapping [111]. In a prospective study involving 24 patients with tumors located near the primary motor cortex, Tarapore et al. reported a high correlation between MEG and nTMS [112]. Nonetheless, the low availability due to the high costs of this method is still a limiting factor for its clinical use. MEG has been also used for predicting postoperative neurological outcome in patients with brain gliomas. Patients with increased functional connectivity presented a higher rate of postoperative deficits at 1 week and 6 months than those with decreased functional connectivity [113].

The clinical importance of DTI-FT in the surgical treatment of brain tumors was demonstrated in many previous studies [114,115,116,117,118,119,120]. Unlike nTMS, MEG, and fMRI, which are responsible for mapping the brain cortex, DTI-FT has a decisive role in mapping the subcortical white matter (see Figure 6). Bello et al. related that DTI-FT presented a high sensitivity for locating motor and language tracts when compared with intraoperative SCS. Moreover, surgery duration and rate of intraoperative seizures were reduced when utilizing both methods [115]. Another study emphasized the relevance of DTI-FT in the preoperative and intraoperative planning of patients with brain tumors. DTI-FT modified surgical approach in 6 of 37 patients (21%) and defined resection margins in 18 of 37 patients (64%) [120]. Recently, studies have reported promising results when combining DTI-FT and nTMS for cortical and subcortical mapping and surgical planning (Figure 7) [93,119,121].

Although this method has proven to be accurate and reliable, care must be taken when utilizing intraoperatively due to the occurrence of brain shift. The concomitant use of SCS is, therefore, strongly recommended.

DCS and SCS are considered the gold standard for intraoperative mapping and have been shown to increase the safety of surgical procedures and EOR [93,122,123,124,125,126].

A meta-analysis involving 8091 adult patients with glioma demonstrated that individuals who underwent resection with intraoperative stimulation mapping (ISM) had a better neurological outcome and a higher rate of GTR than those without ISM [127]. Another clinical report compared two series of patients who had undergone surgical resection of LGG (with and without ISM) and found similar results regarding neurological outcome and EOR [128].

While brain mapping aims to localize the anatomical areas that are responsible for eloquent functions in order to preserve them during surgery, intraoperative monitoring aims to check the functional status of relevant neuronal pathways (e.g., pyramidal, sensory, and auditory) during neurosurgery.

Previous clinical studies reported that intraoperative monitoring with motor-evoked potentials (MEPs) correlated well with postoperative outcome and influenced the course of surgery in patients with glioma, brain metastasis, and intractable focal epilepsy [124,129,130]. Another study with 73 patients undergoing insular glioma surgery showed similar results [131]. The combined use of sensory-evoked potentials (SEPs) and MEPs has been shown to be feasible, and its use is currently a routine in neurosurgical centers. In addition, SEP phase reversal provides a reliable intraoperative localization of the central sulcus [132]. The use of intraoperative neurophysiology plays an important role in the surgical treatment of GBM concerning quality of resection and survival [133]. A 171-patient series reported that even though intraoperative neurophysiology led to transient motor impairment, it did not affect functional outcome [133].

Seidel et al. recently described a new method of intraoperative subcortical mapping, which is performed synchronously with tumor resection [134]. They reported a 5-year series of patients who underwent surgery for tumors located adjacent to the corticospinal tract (CTS) (distance to CTS < 1 cm) [134]. The authors described a dynamic mapping by integrating the probe at the tip of a new suction device [134]. One-hundred eighty-two patients were operated with this new mapping technique [134]. At 3 months, only three patients had a persisting motor deficit that was caused by direct mechanical injury [134]. Another mapping technique involves the use of monopolar stimulation with the train-of-five technique. In this technique, motor thresholds are used to evaluate a current-to-distance relation [135,136]. This enables the surgeon to adjust the speed of resection when approaching the corticospinal tract [135,137]. Rossi et al. developed an advanced mapping technique used for tumors located within the primary motor cortex or subcortical motor pathways [138]. They used high-frequency stimulation (HF-To5 and HF-To2) in the exposed motor cortex in order to establish the safe entry zone for the corticectomy [138]. In addition, they used the same technique to map the subcortical structures and define when tumor resection should be stopped [138].

Intraoperative language mapping has also been performed through a combination of awake surgery (AS) and DCS with positive results and relative safety [125]. AS combined with DCS, SCS, and intraoperative neuromonitoring (IONM) is considered a very effective tool for the surgical treatment of highly eloquent brain tumors [121,139]. Studies have demonstrated that AS is safe, well tolerated by patients, and may be associated with lower costs than surgery under general anesthesia (GA) [140,141,142]. However, a comprehensive cost-effectiveness analysis of this method has not been performed to date. Gerritsen et al. suggested that surgical resection of GBM under AS is associated with greater EOR and less late minor postoperative complications as compared with craniotomy under GA [143]. See Figure 8.

Another clinical report with 309 consecutive patients who underwent AS for brain tumors located near eloquent cortex (language, motor, and sensory) concluded that negative mapping of eloquent areas provided a safe margin for tumor resection and was associated with a low incidence of postoperative deficits [144]. Nevertheless, they reported that the identification of eloquent areas increased the risk of postoperative deficits, likely indicating the proximity of functional cortex to tumor [144]. Some other studies have reported lower rates of postoperative complications, shorter hospitalizations, and greater rates of GTR [145,146,147].

## 7. Confocal Intraoperative Microscope, Intraoperative Mass Spectrometry and Laser Interstitial Thermal Therapy: Current Trends

Since GBMs are highly infiltrative primary brain tumors, scientists are developing strategies for optimizing their surgical resection at a cellular level. The confocal intraoperative microscope (CIM) emerges as a method that provides microscopic images of tissues in vivo during surgical procedures, thus improving the resection of tumor margins [148,149].

In 2010, Sankar et al. presented their initial experience with a handheld, miniaturized confocal microscope in a murine brain tumor model [150]. They reported that the method may help to avoid sampling error during the biopsy of heterogeneous glial tumors and may assist surgeons in detecting infiltrative tumor margins during surgery [150]. Furthermore, Sanai et al. described the results of their clinical study with 33 patients with suspected brain tumors. All patients received intravenous sodium fluorescein, and optical biopsies were obtained within each tumor and along the tumor margins [151]. Afterwards, corresponding pathologic specimens were collected and compared by a neuropathologist in order to identify the concordance for tumor histology, grade, and margins [151]. They described that among HGGs, vascular proliferation, as well as tumor margins were identifiable using CIM [151]. Another study detailed the experience with CIM in 50 microsurgical tumor resections. A concordance rate of 92.9% with the blinded histopathological analysis was reported [152]. Scientists are now working on strategies to automatize the tissue differentiation through an algorithm based on the machine learning concept in order to promote a rapid and reliable categorization of the histopathological findings during CIM-assisted surgery [153].

Mass spectroscopy is a method that provides the differentiation between normal tissue and tumor based on their molecular profile. It is well known that normal tissue and cancer present different molecular profiles, not only quantitatively, but also qualitatively. Agar et al. introduced the integration of desorption electrospray ionization mass spectrometry (DESI-MS) for in vivo molecular tissue characterization and intraoperative definition of tumor boundaries [154]. They used the DESI-MS in a patient with recurrent left frontal oligodendroglioma, World Health Organization grade II with chromosome 1p19q codeletion and reported that the measured data showed an association between lipid constitution and tumor cell prevalence. Another advantage that they reported was the fact that patients do not have to receive systemic injection of any agents [154]. Pirro et al. described the results for 73 biopsies from 10 surgical resections and detailed that DESI-MS allows the detection of glioma and the estimation of high tumor cell percentage at surgical margins with 93% sensitivity and 83% specificity [155]. In addition, the study reported that more than 50% of unresected tumor was found in one-half of the margin biopsy smears, even when postoperative MRI suggested GTR [155]. Another study showed that DESI-MS was able to detect the tumor metabolite 2-hydroxyglutarate (2-HG) from tissue samples of gliomas [156]. Moreover, DESI-MS has also identified isocitrate dehydrogenase 1 mutant tumors with high sensitivity and specificity within minutes, thus providing diagnostic, prognostic, and predictive information [156].

The principle underlying laser-induced interstitial thermotherapy (LITT) is the cytoreduction of the tumor tissue by local thermocoagulation [157]. Carpentier et al. investigated the effects of LITT in four patients with recurrent GBM who were ineligible for a second resection. They inserted a fiber-optic applicator within the tumor, and LITT was performed under continuous magnetic resonance thermal imaging (MRTI). They reported that the procedure was well tolerated with no deterioration of neurological function. Moreover, postoperative MRI showed a decrease in tumor sizes [158]. A multicenter study retrospectively investigated 34 patients with difficult-to-access HGGs who underwent LITT. The 1-year estimate of OS was 68 ± 9% and median PFS was 5.1 months [159]. LITT is a minimally invasive treatment modality for patients with brain tumors that showed promising results in clinical studies and is associated with decreased morbidity [159].

## 8. Raman Spectroscopy and Optical Coherence Tomography: The Future of Glioblastoma Surgery?

Raman spectroscopy is an experimental method that provides a biochemical signature of a tissue, with the potential to provide intraoperative identification of tumor margins [160]. Jermyn et al. developed a handheld contact Raman spectroscopy probe for intraoperative detection of brain tumors. They reported a sensitivity of 93% and specificity of 91% in differentiating normal brain from dense cancer and normal brain invaded by tumor cells [161]. The study concluded that this technology may be able to classify cell populations in real time, thus guiding surgical resection and helping the decision-making process [161]. In addition, other experimental studies with frozen samples demonstrated similar results [160,162]. Neidert et al. established a workflow for stimulated Raman histology (SRH) serving as an intraoperative diagnostic tool in the neurosurgical operating theater of a large European neurosurgical center (see Figure 9) [163]. They reported an easy implementation of the new experimental method into the workflow of neurosurgical resection of tumors [163]. However, certain prerequisites related to the acquisition of tissue samples, data processing, and interpretation must be considered [163]. In a subsequent work of this group, Strähle et al. quantified the neuropathological interpretability of SRH in a routine clinical setting [164]. They performed SRH on 117 samples from 73 cases of brain and spine tumors. The authors reported an accuracy of neuropathological diagnosis based on SRH images of 87.7%, which was not inferior to the current standard of fast-frozen hematoxylin-eosin-stained sections (88.9%) [164]. The rise in the use of Raman spectroscopy is highly interconnected with the development of machine learning models and techniques. A study has reported results of a machine learning model with an ability to discriminate cancer cells from healthy cells with an accuracy of 92.5% [165]. Although this method has shown promising results in clinical and experimental studies, further investigation is required in order to establish the real usefulness of this strategy.

Optical coherence tomography (OCT) is an optical imaging technique that functions as an optical biopsy and provides images of tissues in real time, without the need for the excision and processing of the samples [166].

In 1998, Boppart et al. evaluated the feasibility of using OCT as a real-time intraoperative technique to identify an intracortical melanoma. They used a cadaveric human cortex with metastatic melanoma in this study and reported an effective differentiation between normal cortex and tumor tissue [167]. Another experimental study evaluated the role of OCT in differentiating normal brain and tumor tissue in a mouse model and human biopsy specimens. It was concluded that OCT was able to identify and differentiate those tissues in both murine and human models [168]. In addition, Kut et al. obtained fresh ex vivo human brain tissues from 32 patients with grade II-IV brain cancer and 5 patients with non-cancer pathologies [169]. The study demonstrated that pathologically confirmed brain cancer had lower optical attenuation values at both the solid tumor and infiltrated zones when compared with non-cancer white matter [169]. Afterwards, they used the attenuation threshold (5.5 mm^−1^) to confirm the intraoperative feasibility of performing OCT-guided surgery in a murine model [169]. The study concluded that OCT was able to differentiate cancer from non-cancer tissue [169]. OCT is an emerging optical technology that may help neurosurgeons achieve maximum safe resection of GBM in the near future. Its potential benefits were demonstrated in small experimental and clinical studies. Consequently, high-quality studies on this topic are desirable.

## 9. Conclusions

GBM is a devastating disease with a poor prognosis despite advances in recent years. The combination of surgery and standard radiochemotherapy represents the optimal treatment for combating this lethal condition. Regarding surgical resection of GBM, maximum safe resection represents the golden rule of this treatment modality. Surgical adjuncts such as fluorescence-guided surgery, iMRI, IOUS, IORT, brain mapping techniques, CIM, IMS, LITT, RS, and OCT may be used when properly indicated in order to improve the survival and quality of life of these patients. Many clinical and experimental studies have been published in recent years, and new strategies are released on a frequent basis. However, high-quality evidence is still warranted for most of these new techniques.

## Figures and Tables

**Figure 1 jcm-11-05354-f001:**
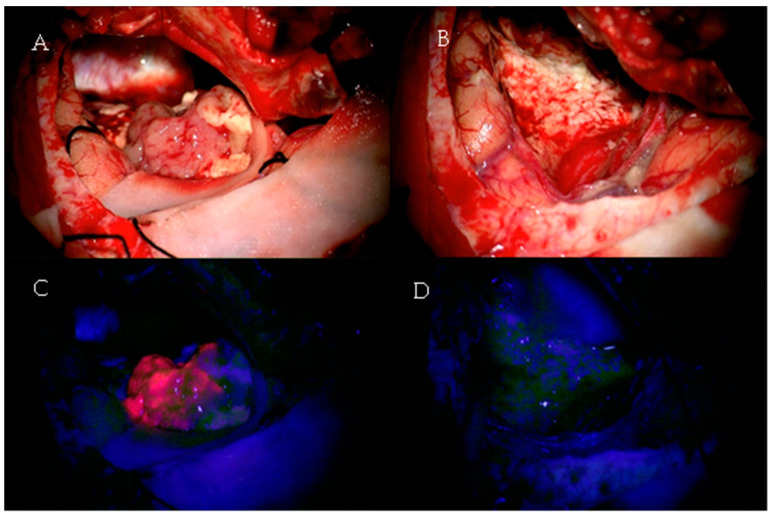
A 61-year-old patient with a left temporal glioblastoma. The upper row shows the operative field in white light prior to (**A**) and after (**B**) tumor resection. The lower row shows the corresponding fluorescence images (**C**,**D**).

**Figure 2 jcm-11-05354-f002:**
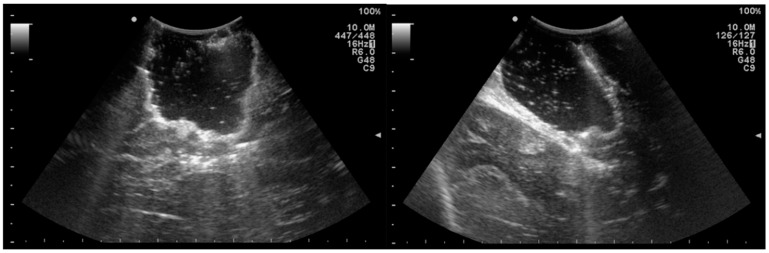
Use of intraoperative ultrasound in a 22-year-old patient with glioblastoma. Axial view (**left**) and coronal view (**right**).

**Figure 3 jcm-11-05354-f003:**
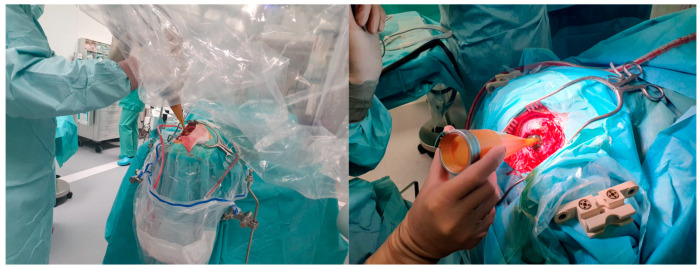
IORT applicator in the surgical resection cavity after resection of a brain tumor.

**Figure 4 jcm-11-05354-f004:**
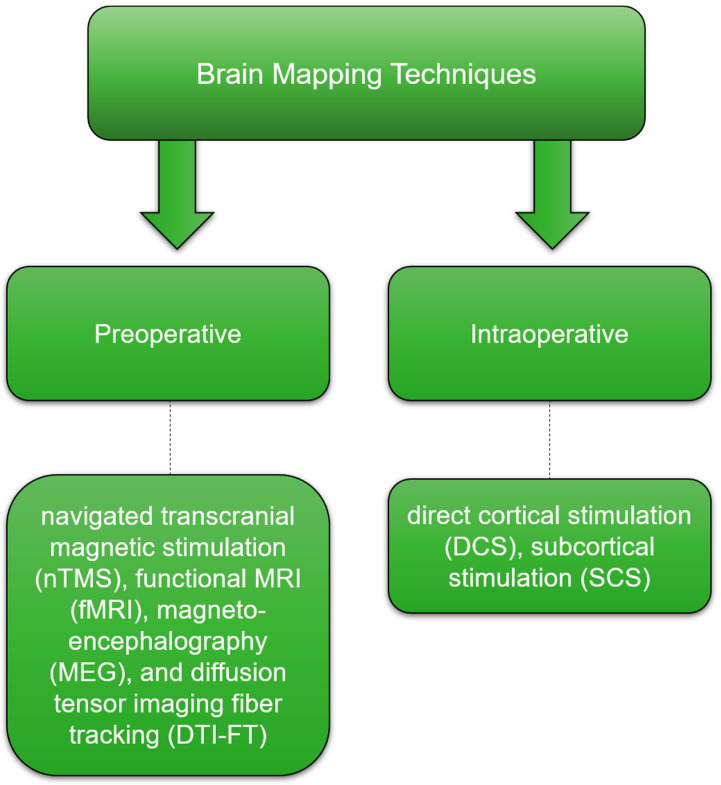
Brain mapping techniques.

**Figure 5 jcm-11-05354-f005:**
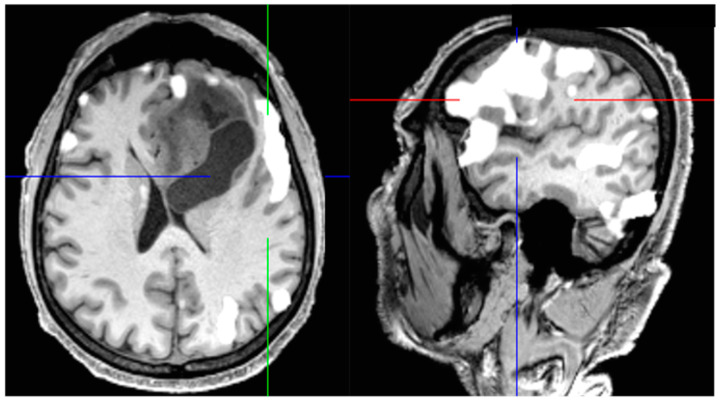
fMRI language task in a patient with high-grade glioma shows Broca’s area clearly lateralized to the left side.

**Figure 6 jcm-11-05354-f006:**
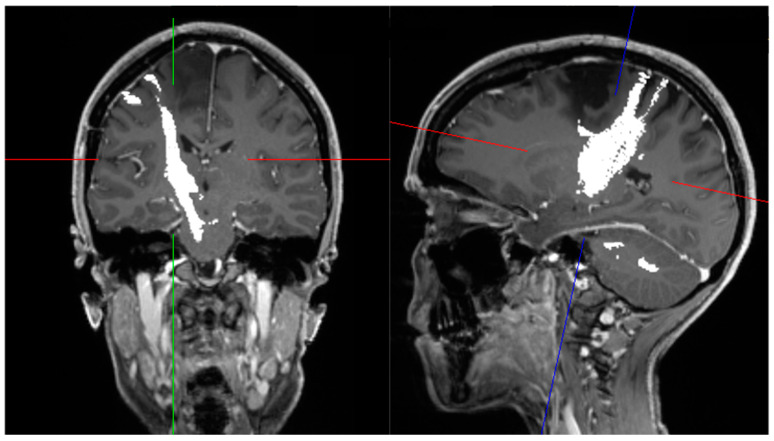
Diffusion tensor imaging fiber tracking showing the right pyramidal tract of a patient with precentral brain tumor. Coronal view (**left**); sagittal view (**right**).

**Figure 7 jcm-11-05354-f007:**
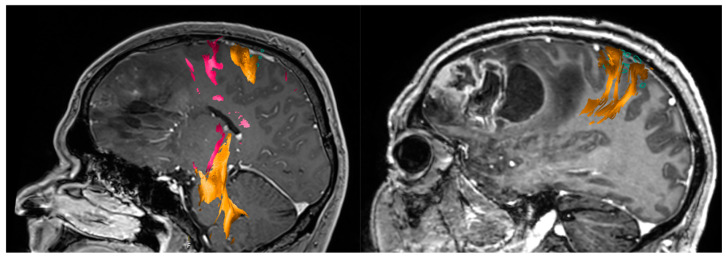
Tractography based on nTMS is a useful adjunct tool in the surgical treatment of glioblastoma.

**Figure 8 jcm-11-05354-f008:**
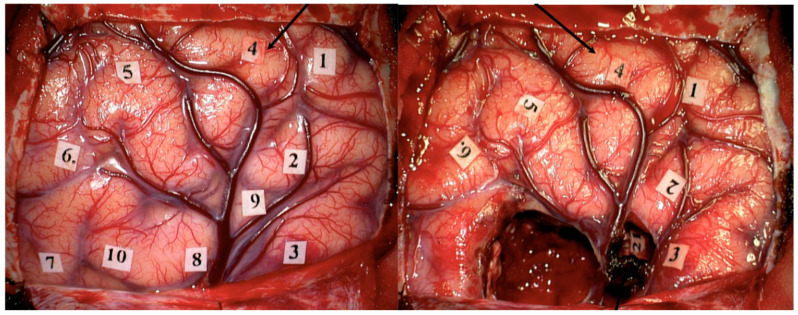
Language mapping during awake surgery in a patient with GBM. The black arrow shows the area related to speech arrest (1.5 mA), before (**left**) and after (**right**) tumor resection.

**Figure 9 jcm-11-05354-f009:**
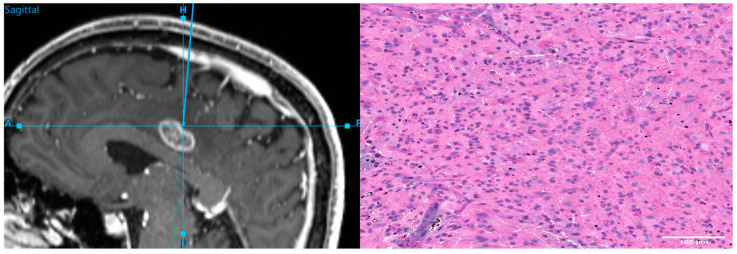
Intraoperative navigation during resection of a brain tumor (**left**) and the corresponding stimulated Raman histology (**right**).

**Table 1 jcm-11-05354-t001:** Main studies evaluating EOR and RV in patients with GBM.

Author, Year	Study Type	Patient Population	Conclusion
Lacroix et al., 2001 [8]	Retrospective	416 patients with GBM	EOR ≥ 98% improved median survival
Sanai et al., 2011 [10]	Retrospective	500 patients with newly diagnosed GBM	EOR ≥ 78% improved OS
Oppenlander et al., 2014 [11]	Retrospective	170 patients with recurrent GBM	EOR ≥ 80% improved OS
Bloch et al., 2012 [12]	Retrospective	107 patients with recurrent GBM	GTR improved OS regardless of initial EOR
Lu et al., 2019 [13]	Meta-analysis	1507 patients with GBM	Maximal resection at reoperation improved OS
Chaichana et al., 2014 [14]	Retrospective	84 patients with newly diagnosed GBM	RV < 2 cm^3^ and EOR > 95% presented the greatest reduction in the risk of death
Orringer et al., 2012 [9]	Retrospective	46 patients with GBM	EOR ≥ 90% improved 1-year survival
Li et al., 2017 [15]	Meta-analysis	1618 patients with GBM	GTR improved 1-year OS and PFS
Grabowski et al., 2014 [16]	Retrospective	128 patients with newly diagnosed GBM	RV < 2 cm^3^ and EOR > 98% improved OS
Bette et al., 2018 [17]	Retrospective	209 patients with newly diagnosed GBM	RV was significantly associated with survival
Chaichana et al., 2013 [18]	Retrospective	259 patients with newly diagnosed GBM	RV < 5 cm^3^ and EOR > 70% improved OS and PFS
Woo et al., 2019 [19]	Retrospective (multicenter cohort)	147 patients with newly diagnosed GBM	MGMT methylation and RV < 3.5 cc improved OS. (EOR was not an independent prognostic factor)
Xing et al., 2018 [20]	Retrospective	292 patients with newly diagnosed GBM	RV, but not EOR, was associated with survival
Pessina et al., 2016 [21]	Retrospective	64 patients with recurrent GBM	RV, but not EOR, was associated with OS and PFS in a multivariate analysis
Sales et al., 2019 [22]	Retrospective	126 patients with newly diagnosed MGMT-unmethylated GBM	RV, but not GTR, improved OS
Esquenazi et al., 2017 [23]	Retrospective	86 patients with newly diagnosed GBM	GTR and near-total resection improved OS
Kreth et al., 2013 [24]	Retrospective	345 patients with newly diagnosed GBM	GTR improved OS; patients who received STR did not show a better OS than those who received biopsy only
Shah et al., 2020 [25]	Retrospective	69 patients with non-eloquent GBM	Supramaximal resection improved OS and PFS compared to matched controls (propensity-matched analysis)

GBM: glioblastoma; EOR: extent of resection; OS: overall survival; GTR: gross total resection. RV: residual tumor volume; PSF: progression-free survival; MGMT: O-6-methylguanine-DNA-methyltransferase; STR: subtotal resection.

**Table 2 jcm-11-05354-t002:** Main studies evaluating the use of 5-ALA in patients with malignant gliomas.

Author, Year	Study Type	Patient Population	Conclusion
Stummer et al., 2006 [27]	Multicenter, randomized, controlled trial	322 patients with suspected malignant glioma	5-ALA group: higher rate of GTR and higher 6-month PFS
Díez Valle et al., 2010 [28]	Prospective	36 patients with GBM	GTR achieved in 83% of patients, EOR > 98% in 100% of cases, and mean EOR was 99.8%
Eljamel, 2015 [29]	Meta-analysis	565 patients with GBM	GTR rate of 75.4% and mean OS gain of 6.2 months
Aldave et al., 2013 [30]	Retrospective	118 patients with HGG	GTR + no residual fluorescence improved OS
Stummer et al., 2000 [31]	Prospective	52 patients with GBM	GTR achieved in 63% of patients; residual fluorescence was a significant prognostic factor
Panciani et al., 2012 [32]	Multicenter, prospective, study	23 patients with suspected HGG	5-ALA-guided surgery showed a sensitivity of 91.1% and a specificity of 89.4%
Stummer et al., 2011 [33]	Randomized, controlled trial *	349 patients with malignant glioma	5-ALA improved OS and 6-month PFS *

5-ALA: 5-aminolevulinic acid; GBM: glioblastoma; GTR: gross total resection; HGG: high-grade glioma; OS: overall survival; PSF: progression-free survival. * Post hoc analysis of the original RCT.

**Table 3 jcm-11-05354-t003:** Main studies evaluating the use of iMRI in patients with malignant gliomas.

Author, Year	Study Type	Patient Population	Conclusion
Senft et al., 2011 [54]	Randomized controlled trial	58 patients with contrast-enhanced gliomas	GTR rate 96% in the iMRI group vs. 68% in the control group
Wu et al., 2014 [52]	Randomized, triple-blind, controlled trial	87 patients with malignant gliomas	iMRI group: Trend toward improved 6-month PFS and higher rate of GTR
Schatlo et al., 2015 [55]	Retrospective	200 patients with HGG	iMRI had no impact on OS
Kuhnt et al., 2011 [56]	Retrospective	153 patients with GBM	iMRI contributed to optimal EOR with low postoperative morbidity
Corburger et al., 2017 [57]	Prospective	170 patients with GBM	Surgery with iMRI presented higher OS and lower complication rates than previously published data
Kubben et al., 2014 [58]	Randomized, controlled trial (interim analysis)	14 patients with suspected GBM	iMRI group: no advantage with respect to EOR, clinical performance, and survival
Marongiu et al., 2016 [59]	Retrospective	114 patients with GBM	iMRI improved both EOR and 6-month PFS
Li et al., 2017 [60]	Meta-analysis	Patients with glioma	iMRI improved rate of GTR and 6-month PFS

iMRI: intraoperative magnetic resonance image; GTR: gross total resection; PSF: progression-free survival; HGG: high-grade glioma; OS: overall survival; GBM: glioblastoma; EOR: extent of resection.

**Table 4 jcm-11-05354-t004:** Main studies evaluating IOUS in patients with brain tumors.

Author, Year	Study Type	Patient Population	Conclusion
Jakola et al., 2011 [70]	Retrospective	88 patients with glioma	IOUS improved QOL
Saether et al., 2012 [69]	Retrospective	192 patients with GBM	IOUS improved survival since introduced in their department
Moiyadi et al., 2015 [68]	Prospective	88 patients with glioma	Navigable US improved PFS and OS
Mahboob et al., 2016 [71]	Meta-analysis	739 patients with glioma	IOUS improved EOR
Prada et al., 2016 [72]	Prospective	10 patients with GBM	CEUS was extremely specific in identifying residual tumor
Moiraghi et al.,2020 [73]	Retrospective	60 patients with supratentorial gliomas	N-ioUS improved EOR and neurological outcomes

IOUS: intraoperative ultrasound; QOL: quality of life; GBM: glioblastoma; US: ultrasound; PSF: progression-free survival; OS: overall survival; EOR: extent of resection; CEUS: contrast-enhanced ultrasound; N-ioUS: navigated intraoperative ultrasound.

**Table 5 jcm-11-05354-t005:** Main studies evaluating the use of IORT in patients with brain tumors.

Author, Year	Study Type	Patient Population	Conclusion
Sakai et al., 1991 [79]	Prospective	73 patients with malignant glioma	IORT improved median survival
Fujiwara et al., 1995 [80]	Prospective	36 patients with glioma	IORT improved median survival
Schueller et al., 2005 [82]	Retrospective	71 patients with malignant glioma	IORT did not improve OS compared to a historical group
Nemoto et al., 2002 [88]	Retrospective	32 patients with malignant glioma	IORT did not improve survival compared to matched control cases
Giordano et al., 2019 [86]	Clinical trial	15 patients with newly diagnosed glioblastoma	IORT was associated with manageable toxicity
Sarria et al. 2020 [89]	Retrospective	51 patients with glioblastoma	Improved efficacy and safety of IORT with low energy X-rays compared to historical data

IORT: intraoperative radiotherapy; OS: overall survival.

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
