# Peer review of "Surgical Treatment of Glioblastoma: State-of-the-Art and Future Trends"

_jcm, 2022, doi:10.3390/jcm11185354_

Round 1

Reviewer 1 Report

The authors in their review article „Surgical Treatment of Glioblastoma: State of the Art and Future Trends“ described the current surgical treatment of glioblastoma tumors. They present substantial conclusions of  all relevant studies ranging from the extent of tumor resection to the latest, experimental  technical solutions. Although there is a very long list of studies, their paper is not written in a confusing manner. However, some studies quite obviously tried to make conclusions in favor of the medial device industry, but the authors avoided any misinterpretation in the paper.  

Author Response

We sincerely express our deepest gratitude for the positive responses to our manuscript and the concise summary of our work.

Reviewer 2 Report

The manuscript reviewed the current surgical treatment strategies for glioblastomas.

First the review is non-systematic leading to a very large bias in the information presented, which is selected by the authors. Although the authors cover a broad spectrum on the knowledge of surgical treatment strategy for glioblastomas, this article provides no novel information on glioblastomas compared to previously published reviews. Glioblastoma is a far too wide topic to summarize the recent knowledge in such a short comprehensive review

The section of intraoperative radiotherapy: Targeting infiltrative margins is relatively interesting.

Author Response

Thank you for this comment. We totally agree that non-systematic reviews are more susceptible to selection bias. However, in this paper we have tried to present not only the agreements but also the controversies in each topic. The studies were presented in an impartial way without revealing our individual preferences as experts in the field. We have summarized the methodology of each cited study (e.g. study design- prospective, clinical trial or retrospective study?-number of participants, p value, confidence interval) in order to give the readers the necessary tools to judge the level of evidence. In addition, we have cited many studies that  showed different conclusions in order to overcome this selection bias and to provide as much transparency as possible. Moreover, we have stated in the abstract as well as in the introduction section that this paper is a comprehensive review in order to demonstrate our commitment with transparency. We think that systematic reviews play an important role in answering specific scientific questions, which was not the primary objective of this paper. The aim of this paper was to present a comprehensive update on the most relevant issues regarding the surgical treatment of glioblastoma. Our decision to submit the paper in the Journal of Clinical Medicine reflects our objective to provide colleagues from other specialties with an overview of this important topic, demonstrating the state of the art and the future trends in the field. 

Reviewer 3 Report

Surgical resection combined with adjuvant radiochemotherapy remains the main therapeutic option for glioblastomas (GBM). Maximum safe resection remains the main objective when surgically approaching these patients due to its association with longer survival.

In the present article, Arthur H.A. Sales 
et al. offer a comprehensive literature review on the surgical adjuncts that may be used in the surgical treatment of GBM.

First paragraph summarizes well the agreements and controversies on the Extent of Resection (EOR) and Residual Tumour Volume (RV). The more the better, provided that no new or worsen deficit would occur postoperatively.

Second paragraph deals with fluorescence guidance. The authors show with great enthusiasm its role in detecting tumour infiltration at the cavity margins.

Third paragraph emphasises the facts that to date ioMRI has no clear benefit and its routine use should not be recommended in the context of glioblastomas.

Forth paragraph explains that intraoperative ultrasound is easily accessible but highly depended on the operator. There is again low level of evidence.

Fifth paragraph concerns intraoperative RT. Studies analysed are quite old studies. Probably there has been more interest recently in dose escalation based on tumour heterogeneity apprehended by image targeting (ie, multimodal MR imaging and new PET tracers).

Sixth paragraph and probably the most important emphasizes the major role of brain mapping techniques. Distinction has been made between intraoperative and preoperative non-invasive techniques, the former being the more reliable and the “gold standard” to date.

Last two paragraphs reports preliminary data using confocal intraoperative microscope, intraoperative mass spectrometry, and laser interstitial thermal therapy on the one side and Raman spectroscopy and optical coherence tomography, on the other side. It is obvious that data are very preliminary. There could be an interest in the future.

There has been a lot of similar reviews on the topic. In our opinion, this paper would be more valuable if enriched by expert opinion. Indeed, the hierarchy between what should be necessary and what is very "incidental" is not that much striking for non-specialists. We would then suggest the author to add comments on their own view and/or practice, particularly regarding combination of these tools. Which combination(s) would they push forward? Why? In the same line, we would also have liked the author to express their views on the concept of “functional neuro-oncology” applied to glioblastoma (ie, supra-total resection with functional limits). There have been some controversies in the literature. A few words of ethics and economics would also have been welcome.

Author Response

Thank you for this valuable comment. With respect to the IORT section: We think this is an excellent suggestion. Therefore, we have included a few recent studies, including a study regarding dosimetric data in this section. Regarding comments on our own view: we think this is an interesting suggestion. However, since this is not a systematic review, we have tried to present the studies in an impartial way without revealing our individual preferences in order to minimize the selection bias and improve the quality of our work. In addition, the authors represent two different institutions with different approaches to these patients. Therefore, we have decided to present the studies impartially, always summarizing the methodology of the presented studies in order to give the readers the necessary tools to judge the level of evidence and applicability.

With respect to the concept of functional neuro-oncology: Thank you for pointing this out. We have included a few words in the section on Brain mapping and monitoring strategies regarding this issue.

Reviewer 4 Report

The authors furnished a clear up-to date overview of the surgery for glioblastomas. The article is well organized in paragraphs reflecting the most important issues on the matter.

In particular, the authors analyzed the issues concerning the role of extent of resection and residual volume, and the importance of some technical intraoperative innovations as fluorescence, MRI, ultrasound, radiotherapy, brain mapping and monitoring strategy, confocal microscopy, mass spectrometry and laser interstitial thermal therapy.

From their analysis, some conclusions might be driven. For instance: 1. Residual tumor volume has a more prognostic value than extent of resection; 2. Technical intraoperative innovations still lack high quality evidences regarding their role in improving surgical results from the oncological point of view.

Summarize and disclose the achievements in malignant brain tumor surgery is valuable and the authors carried out this commitment thoroughly.

P.S. Legend of figure 10 should be corrected (Left/Right)

Author Response

Thank you for this comment. We sincerely appreciate the concise summary of our work. Figure 10 has been removed of this version of the manuscript due to copyright issues.

Round 2

Reviewer 2 Report

Glioblastoma is a far too wide topic to summarize the recent knowledge in such a short comprehensive review.

Author Response

-